# Exploration of Social Exclusion among Older Chinese Immigrants in the USA

**DOI:** 10.3390/ijerph20032539

**Published:** 2023-01-31

**Authors:** Ling Xu, Jia Li, Weiyu Mao, Iris Chi

**Affiliations:** 1School of Social Work, University of Texas at Arlington, 501 W Mitchell St., Arlington, TX 76019, USA; 2Department of Social Work, Faculty of Social Science, The Chinese University of Hong Kong, Hong Kong, China; 3School of Social Work, University of Nevada, Reno. 1664 N. Virginia Street, Reno, NV 89557, USA; 4Suzanne Dworak-Peck School of Social Work, University of Southern California, Los Angeles, CA 90089, USA

**Keywords:** accumulating disadvantages, older Chinese immigrant, social exclusion, socio-cultural environment

## Abstract

Background: Older adults are vulnerable to social exclusion and its detrimental health outcomes. However, few existing studies focus on the social exclusion of older adults as immigrants and ethnic-minority individuals. To fill the gap, drawing on the conceptual framework of old-age vulnerability, this study explored the multi-dimensional experience of social exclusion for older Chinese immigrants in the U.S. and investigated how old age and immigration exacerbated older adults’ experiences of social exclusion. Method: The study included 24 in-depth face-to-face individual interviews in Los Angeles and employed thematic analysis on the qualitative data. Themes were identified until consensus was reached among the research-team members. Triangulation of multiple analysts was used to avoid analytic bias. Findings: Findings showed that older Chinese immigrants experienced social exclusion in the following dimensions: basic services; material and financial resources; social relations and activities; socio-cultural aspects; and neighborhood/territory/community. In addition to age-related factors, immigration-related factors, including changes in physical and socio-cultural environments and legal status, also contributed to older immigrants’ extra vulnerabilities to social exclusion. Conclusions: This study provides useful information and strategies for human and healthcare service providers to find ways to overcome social exclusion and enhance older Chinese immigrants’ social inclusion in the U.S.

## 1. Introduction

In many ways, older adults are more vulnerable than other groups in society due to diminished functional health and reduced social activities [1]. The vulnerability is even more evident among older immigrants who may undergo acculturation stress or disrupted family and social relationships [2,3]. Social exclusion is one important concept that focuses on the vulnerability of individuals and groups in society and has been used widely across different disciplines in the social sciences [4], but largely neglected in the health field. While most of the literature on social exclusion is concentrated on general older adult populations, less is known about the social exclusion of older immigrants, especially older Chinese immigrants. Research demonstrates that older Chinese immigrants in the U.S. have reported more mental health problems than other ethnic-minority older adult groups [5,6,7,8,9]. Additionally, current contemporary events have increased anti-Asian (especially anti-Chinese) bias and attacks on Asian seniors in the U.S. [10,11]. Thus, it is very necessary to understand the level of social exclusion of this population. By exploring the social exclusion experience of older Chinese immigrants and explaining the risk factors with qualitative interviews, this study will help understand the current social exclusion or anti-Asian incidence/crime experienced by Asian immigrants, despite the age of the data used in the present study, which were collected in 2010, and will help health or human service providers promote older immigrants’ quality of life and integration into society.

### 1.1. Defining Social Exclusion

The concept of social exclusion was first implemented in France in the 1960s [12], and it is commonly used to refer to the process that leads to a breakdown of the relationship between the individual and society [13]. Social exclusion is a dynamic process that prevents people from accessing different elements of the social, economic, political, or cultural components of everyday life [14,15]. Although there is no widely accepted agreement on how to measure social exclusion and which potential elements should be included [16], studies have shown that social exclusion constitutes multidimensional contents, involves deprivation across a range of domains, and can be referred to as a process, outcome, concept, experience, or even a feeling [17]. Because social exclusion is a multidimensional concept, it allows for a rich understanding of how an individual, and in turn a population, can experience exclusion across many facets of life [4].

Walsh and colleagues conducted a two-stage scoping review and summarized six domains of social exclusion experienced by older people, specifically [17]. The six domains included: (1) neighborhood and community (2) services, amenities and mobility; (3) social relations; (4) material and financial resources; (5) socio-cultural aspects; (6) civic participation [17]. This framework provided a comprehensive guideline for summarizing and categorizing older people’s experiences in social exclusion in this paper.

### 1.2. Social Exclusion among Older Immigrants

Social exclusion is always expressed or understood in a specific context [17,18]. Ethnicity is known to be predictive of social exclusion; minority immigrants are more likely to be excluded from society than ethnic-majority populations, as they suffer from certain types of discrimination, unemployment, poverty, differential access to housing, and limited political and social participation [16,19,20]. Empirical studies have examined what social exclusion immigrants in general experience. For example, Cavalcanti [21] reported that immigrants residing in Europe experience economic, social, and cultural dimensions of social exclusion that make them different from the native population. Young immigrants in the U.S. generally experience educational exclusion, labor-market exclusion, spatial exclusion, relational exclusion, and finally, socio-political exclusion [22]. African immigrants (aged 18–62) in the United States experience four dimensions of social, cultural, structural, and economic social exclusion [23,24], or experience social exclusion in the form of material deprivation, limited access to basic social rights, limited social participation, and insufficient cultural integration [25].

However, the construct of social exclusion remains largely unknown in the context of aging immigrant populations, with most research focusing on social exclusion among the general older populations or immigrants. The social exclusion of older and immigrant people is a particularly pressing issue, due to intensive demographic changes in an individual’s family, network, neighborhood, and social environment after immigration [26]. For example, older Chinese immigrants experience challenges in family and social relationships, as well as support from local communities due to limited language proficiency, lack of transportation, and economic constraints [27,28]. They also tend to have smaller friendship networks [29] and social networks [30], and experience acculturation-related conflict with children after immigration [31,32].

Limited literature has shown that amongst older Korean American immigrants, three dimensions of social exclusion significantly influence participants living in poverty: exclusion from social and civic engagement; exclusion from asset building, with no homeownership or retirement-pension ownership; and exclusion from the labor market, with limited work opportunities and income from social security [33]. However, there have been no specific studies that have explored the experience of multidimensional social exclusion among Chinese immigrant older adults in the U.S. Chinese immigrants are a large population, with more than 12 million residing worldwide outside of China [34]. Their experiences can be unique, and cannot be generalized from Korean or other immigrants. Though it is established that older Chinese immigrants in the U.S. are a rapidly-increasing population at high risk of mental health problems such as more anxiety symptoms [8], a heightened sense of loneliness [35], similar or higher levels of stress or depression than the general older population in the U.S. [5], and higher levels of suicidal ideation than their counterparts in China [7], we know little about this population’s particular vulnerability to social exclusion. Learning what social exclusion this population experiences and what factors contribute to their social exclusion in the U.S. may help policymakers and program administrators to avoid such factors in order to increase their social inclusion in the U.S.

### 1.3. Risk Factors of Social Exclusion: ‘Old-Age Vulnerability’ Framework

While the six-domain framework can be borrowed to understand the nature of immigrants’ experiences of social exclusion, understanding the risk factors of social exclusion could help health or human service providers be aware of these harmful factors and help older immigrants be less likely to be excluded from society. To better understand the accumulating disadvantages of older immigrants, we used the old-age vulnerability framework of Schröder-Butterfill and Marianti [36] to guide the design of this study. This framework illustrates a potential pathway from interactions among the domains of exposure, threats, and coping capacities, to bad outcomes. “Bad outcomes” for older adults generally include untimely or degrading death, lack of physical care and health care, oversupply of care and interference, poverty, exclusion from society, homelessness, loss of autonomy and dependence, institutionalization, lack of social contacts and loneliness [36]. “Exposure” refers to the susceptibility or potential risk factor of encountering a given threat or outcome. For instance, entering old age is a risk factor for social exclusion, and exposure takes place before bad outcomes. “Threats” refer to specific events that have the power to propel people towards bad outcomes. Immigration to a new country in the present study is considered a threat. “Coping capacity” refers to the resources that can protect people from bad outcomes, including individual capacities, social networks, and formal social protection. Schröder-Butterfill and Marianti [36] noted that the four domains may be closely interlinked with each other. Exposure, threats, and coping can compensate for each other or be mutually exacerbating in the process of leading to bad outcomes. This theoretical framework is suitable to guide the present study exploring the bad outcomes of social exclusion of Chinese older immigrants who are exposed to the aging process and faced possible threats of immigration-related challenges.

As older adults, they may first be affected by vulnerability related to age. As people grow old, the chance that they become more socially excluded is greater than when younger [37]. Being old gives people less access to services such as public transport and social activities, due to decreased physical health or disability [14,16,38]. Aging makes people more vulnerable to poverty, due to loss of income after leaving the labor market [38]. It also makes them more likely to live in an undesirable neighborhood [4,38]. Finally, this population often experiences age-based discrimination [16].

Immigration, together with aging, often results in “double jeopardy” for the population of older Chinese immigrants to the U.S. Moving to a new country creates significant changes in older adults’ surrounding environment. They not only have to adapt to an unfamiliar physical environment, but may also face barriers to integrating into the social and cultural environment of the destination country. They, therefore, are more vulnerable to social exclusion than general older adults, due to the emerging threats from unfamiliar physical, social and cultural environments, especially when the coping capacity of older immigrants themselves, their families, and their host society cannot mitigate these threats. A quantitative study among Korean immigrants in the U.S. suggested that in addition to age-related factors (e.g., limited working opportunities), immigration-related factors, such as linguistic isolation, U.S. citizenship-status restrictions, and short duration of residency in the U.S., also contributed to their experience of exclusion in social and civic engagement [33]. One qualitative study found that hostile attitudes, deficient language-acquisition, and a longing for the former home-country affected feelings of social and emotional exclusion for older immigrants, from the former Soviet Union to Finland [39].

To the best of our knowledge, no study has been conducted on what factors contribute to older Chinese immigrants’ social exclusion experience. Older Chinese immigrants generally experience high levels of resource deprivation, low levels of acculturation, increased language barriers, and elevated social isolation, compared to other ethnic-minority older adults in the U.S. [27,40]. They also experience some discrepancy between expectation and receipt of filial piety and family support from adult children in the process of immigration [27,41]. Therefore, it is highly likely that older Chinese immigrants may have multiple risk factors for social exclusion that arise from the aging process itself and the threats they face after immigration. However, Chinese immigrants often attract less research attention, in part due to the “model minority” myth [42]. More research is needed into the social exclusion experience of this population. By focusing on the Chinese population, the largest ethnic group in the world that reports high vulnerabilities, this study will shed important light on the nature of ethnic aging.

### 1.4. Aims and Objectives of the Present Study

The present study aimed to fill two research gaps in the literature about social exclusion among older adults. First, even though there has been literature discussing latelife or immigration-related social exclusion, the specific patterns of multi-dimensional exclusion experienced by particular ethnic-minority older-adult groups remain largely unknown. This study focuses on older Chinese immigrants aged 65 and above, a group that has been under-investigated in the literature despite the vulnerabilities mentioned above. Second, and more importantly, most existing literature only focuses on multiple domains of social exclusion as predictors of well-being [20,24]; the risk factors or reasons that social exclusion is exacerbated for minority older adults have been largely ignored.

This study specifically addressed the following two research questions: (1) What are the experiences of “social exclusion” from the perspective of older Chinese immigrants in the U.S.? (2) Drawing on the conceptual framework of old-age vulnerability [36], what are the risk factors, immigration-related factors in particular, that formulate older Chinese immigrants’ experience of social exclusion in the U.S.?

## 2. Materials and Methods

### 2.1. Participants

This study employed an in-depth individual interview method to explore social exclusion and its risk factors among older Chinese adults in greater Los Angeles, the second-largest metropolitan area in the U.S. that has Chinese Americans. Older adults aged 65+, self-identified as Chinese American, and living in the L.A. area were eligible to participate. Convenience quota-sampling was used for interviews. The research team first developed a sampling strategy based on the background information collected during recruitment, such as gender, age, educational level, language, and religion, to avoid selection bias and balance the samples to generate diversified insights from participants. Respondents were recruited from areas with a high concentration of individuals of Chinese descent, where the Chinese culture (e.g., Chinese stores, groceries, and restaurants) is apparent. Recruitment occurred through the distribution of flyers, word of mouth, and the help of administrators in organizations including senior housing facilities, senior centers, adult day health care facilities, and social services agencies. More detailed information about recruitment can be found elsewhere [41,43].

Sampling and interviewing were stopped when data reached saturation, with no additional data found to develop new categories or themes. The final participant roster was comprised of 24 Chinese immigrants aged 65 or above who had immigrated to the United States from China, including mainland China, Hong Kong, and Taiwan, and were residing in the greater Los Angeles area. A summary of additional background information collected about the participants is shown in Table 1.

### 2.2. Procedures

Before the interview, the interviewers attended a two-session intensive training on qualitative interviews offered by the researchers. The interviews were conducted in 2010 in various private or quiet places that the participants and researchers had agreed on, such as meeting rooms in the local library, conference rooms within an agency, a nearby park, etc. All interviews were conducted in Chinese (either Mandarin or Cantonese) and in a one-on-one manner by trained interviewers who were bilingual, with the ability to speak Mandarin or Cantonese fluently. The average interview session time was approximately 70 min. All interviews were audio recorded. In addition, every participant received a gift card as compensation for his or her time.

### 2.3. Ethics

All interviewers and research team members completed the Human Subject Projections training and obtained the certificate before the study started. The study protocol was approved by the Institutional Review Board (IRB) of the University of Southern California (Approval # APP-09-08831) prior to conducting the study. Only after the participants signed the consent form did the interview start. Contact information of the principal investigator and IRB office was included in the consent form, and each participant kept a signed copy.

### 2.4. Instrumentation/Materials

To develop an appropriate, accurate, and adequate interview guide for the study, scholarly journals and previous research on related subjects were studied and discussed among research team members. A pilot study was carried out through convenience sampling of four older adults. Amendments were made after the pilot study, and then the research team completed and finalized the interview guide. All interview questions were translated into Chinese (either Mandarin or Cantonese), as appropriate for participants.

The final questionnaire consisted of two major sections: a close-ended survey on basic background information and an open-ended, in-depth individual interview guide. Background information was collected from each individual prior to the interview. The interview (see Appendix A) included a series of guiding questions with structured probes that asked about an individual’s perception and lived experience in the individual, community, neighborhood, and services/resources domains. These guiding questions were designed to be followed by other supporting questions that required the respondent to provide detailed experience or ideas to support his or her stance. 

### 2.5. Data Analysis

All the interviews were audio recorded with the consent of participants. Audio-taped data were transcribed verbatim into Chinese. After each researcher had transcribed independently each audio tape, a second transcriber listened independently for a second transcription. Researchers then compared the first and second transcriptions for disparities. A final transcription was generated after the researchers collectively discussed the most accurate interpretation of any disparities. A back-translation approach was also utilized, which included translation into English and then double-checking for errors by a second, independent researcher to ensure equivalence in both languages, as recommended by Knight, Roosa, and Umaña-Taylor [44].

The researchers utilized thematic analysis and focused on the identification of themes through a coding process that progressed from description to interpretation [45,46]. Thematic-analysis process included: (1) familiarization with the data, (2) generation of initial code, (3) identification of potential themes, (4) reviewing themes, (5) defining and naming themes, and (6) producing the report [45]. The researchers moved iteratively throughout these six phases (e.g., recursive analysis). Analyst triangulation was used to address potential investigator bias, and to enhance the credibility of the study. Using multiple analysts/observers/investigators to review findings can provide a check on selective perception and illuminate blind spots in an interpretive analysis [47]. First, members of the research team coded the data independently, to provide the basis for a preliminary coding framework. The researchers then reconvened to compare coding outcomes, to enhance consistency and expand the dimensions and comprehensiveness of codes. Codes were later combined or synthesized into broader, recurrent themes. After the discussion, researchers eliminated quotations that were irrelevant, checked the accuracy of previous quotations, and made corrections when necessary. The group met a second time to discuss the recoded quotations. There was final agreement on 85% of the codes applied, indicating good reliability [48]. To reduce the likelihood of researcher bias, reflexivity was also considered through several techniques, such as jotting down notes about participants’ comments and researcher’s thoughts during the interview, memoing as soon as possible after an interview, and developing and continually editing the researcher’s subjectivity statement. These processes are not separate from the data analysis process, but embedded into it. Atlas.ti 7 (ATLAS.ti Scientific Software Development GmbH, Berlin, Germany) was used in the coding process.

## 3. Results/Findings

To facilitate the thematic understanding of multidimensional social exclusion, the research team referenced Walsh’s framework. Five dimensions of social exclusion were experienced by the older Chinese-immigrant participants: exclusion related to basic service; material and financial resources; social relations and activities; socio-cultural aspects of society; and neighborhood/territory/community. Moreover, according to the theoretical framework of old-age vulnerability, exposure, risk, and coping capacity work together to result in social exclusion. Therefore, the team can report the experience and its risk factors in each dimension of social exclusion at the same time in the findings.

### 3.1. Exclusion from Basic Services and the Risk Factors

Many participants reported exclusion from basic services including amenities, healthcare, housing, and mobility, etc. Going to church or attending religious activities proved to be important components of immigrants’ daily lives, especially for older people. However, limited physical mobility prevents them from attending religious or cultural activities. For example, P6 (Female, 77) mentioned that she could not go to the traditional Chinese dance club because of her back pain or discomfort. Another participant mentioned that she and her friends could not go to church due to impaired physical ability.

I don’t go to any [religious meetings]. I am old. [The church] wants me to refer my friends, but they won’t go. There is no transportation, they (friends) said their feet are not good anymore and couldn’t walk. There is no point (to go or invite people to church). We are not able to do that. Where could they go? …You are too old to do such things.(P9, Female, 90)

Some participants also reported exclusion from some cultural activities, partially due to financial concerns:

Sometimes they organize some trips, such as visiting some cultural museums or going to famous sites to learn different ethnic and geographic culture. I will go sometimes (if no cost or the cost is low). I won’t go to those that cost a lot of money, such as a trip that costs 1000 dollars to go somewhere for seven days. I want to participate and enjoy the diverse culture in the U.S., but I can’t afford it.(P12, Male, 68)

Services for housing and transportation are important aspects of daily life. Some participants reported that they experienced difficulties receiving sufficient services for housing maintenance in their senior apartment, because they had limited ability to take care of the outdoor spaces of a residence. Moreover, this is exacerbated by the language barrier.

If there is an agent—because people are all old, you know, people like us knowing little English—sometimes it’s better to have a person helping us with this [housing stuff], but there is no such person.(P11, Male, 73)

Sometimes my car would break down. It was troublesome when it happened. Because my English is not good—that is, I didn’t know how to call [for assistance] and I couldn’t find anyone to help. So I stood for very long time and it didn’t get solved until the police came.(P10, Male, 75)

Language barriers can also prevent seniors from using medical services. Participant 20 (male, 68) mentioned that most older adults tried their best to have Chinese doctors. If they have a Western doctor, older Chinese immigrants may not understand what the doctor says, increasing anxiety about their health. An insufficient supply of resources can also contribute to older Chinese immigrants’ social exclusion from community services:

Hmm…computer classes. It was hard to get in but I did finally. I meant, I was on the waitlist for a year. Many people wanted to take that class…I waited for a year.(P17, Female, 65)

There are no community activities during holidays. This is the worst. For example, on Saturdays or holidays, these places are all closed. The older people will be like…they don’t know where to go, because their children are mostly in other states. So, we elders can only stay at home during holidays.(P20, Male, 68)

Immigration status influences participants’ eligibility for using social services. Not holding a U.S. citizenship means that some older Chinese immigrants are excluded from material resources, bearing an extra burden of paying for medical services.

The services provided to green card holders and citizens are very different and unequal. We as green card holders do not have any relief funds, no insurance, we got no services…If you can get the citizenship, it’s easier for you to get medical treatment. People with ‘red and blue’ (i.e., Medicare) cards have priority. If we want something, for example once I wanted a crutch, they said I couldn’t have it if I only had a white (i.e., Medicaid) card.(P1, Male, 86)

### 3.2. Exclusion from Material /Financial Resources and the Risk Factors

Finance or material resources were important aspects of social exclusion in the literature. Some participants reported that they themselves or other seniors experienced material deprivation in their basic living, receiving neither sufficient medical services nor engaging in leisure activities:

When I go out for a walk in the morning, I see Chinese seniors live in difficult or harsh conditions. I walk in the morning, early, before dawn. I see a lot of seniors picking up plastic bottles from trash. There are many of them, not just one or two. Some of them really live in a hard situation.(P2, Female, 74)

The healthcare system in the U.S., so different from China’s, partially contributes to some participants’ social exclusion from material and financial resources, based on the perspectives of the participants:

Many people can’t buy the insurance. Some are not in the age [of 65 or above] for being eligible for the governmental benefits. You may be ill at 64 or 62 but you don’t have the insurance. If you buy the insurance at 65, it will be about 700 dollars per month. How are you gonna pay it? After you pay it, you will have no food.(P20, Male, 74)

I had some concerns about the medical system. After I came here, I needed to fix my teeth. But then I found out I could not have reimbursement here. I spent a lot of money on that. It was way too expensive; something like thousands of dollars. So, I realized I needed to go back to China to fix my teeth, because I have 100% reimbursement in China. So, I’m going back next year to fix my teeth. It’s too expensive here. No one can afford it.(P13, Female, 76)

### 3.3. Exclusion from Social Relations/Activities and the Risk Factors

Social relations and social activities are another dimension where some participants experienced social exclusion. They felt excluded from family relationships, and reported loneliness when their children were absent. This is especially the case for those who do not live with their children:

Of course, it’s better to have friends. I won’t be as lonely as now. Living by oneself is lonely. They (children) all go to work and leave me alone. Actually, I spend most of the time all by myself. They don’t even stay at home during holidays. They’re out all day.(P9, Female, 90)

Cultural differences are reflected in the relationship between adult children and parents. Older Chinese immigrants may maintain more Chinese cultural values, expecting children and parents to be more interdependent with each other. However, their adult children may have acculturated more to the U.S. style of independence and distance from aging parents, which may contribute to loneliness:

Many older people sometimes cannot accept their (current unsatisfactory) situation. This is generational gap. The problems among seniors here is that they don’t know how to adjust to the generational gap. They then behave oddly. The young people will not like it, right?(P19, Female, 76)

[In Taiwan] sons and daughters support their parents. Here [in the US] everyone lives his/her own life…In Taiwan, if you don’t practice filial piety, you will be put in jail.(P4, Male, 92)

Some participants reported a loss of close friends as they grow older. Immigrating to a new country separated them from their friends back in China, and made it even harder to form and maintain new friendships, especially close relationships:

I used to, but very few now. Many passed away. Many moved, they moved with their children…for several of them, I was not happy…I tell them, ‘I am not happy about the change’(P19, Female, 76)

Friends are… my old friends. They are mostly friends I met in Vietnam. Now the new friends are just normal friends. Good friends have known me from a long time ago.(P12, Male, 68)

My…because most of those friends are…those so-called old friends, were in the past, I got to know in the past. Most of them are in Canada or Taiwan or Hong Kong. So we usually, sometimes, nothing special, like Chinese New Year, we give each other a call.(P16, Female, 70)

Language barriers were not limited to English. Any language barrier may cause miscommunication and prevent older Chinese immigrants from developing social relationships with other Chinese immigrants. For example:

I think older people should learn more languages. Like now I have a schoolmate who is always speaking [his or her] own dialect. So even if we want to communicate with [him or her], it is hard because we don’t know [his or her] language. So if you don’t know the language, it is hard. It is hard if you want to help [him or her].(P17, Female, 65)

### 3.4. Exclusion from Socio-Cultural Aspects of Society and the Risk Factors

Social exclusion in socio-cultural aspects refers to the low level of socio-cultural integration of older Chinese immigrants. Language is the basis of communication and social interaction. Limited English language proficiency can influence older Chinese immigrants’ feelings of being excluded from mainstream society. For example, because of limited English proficiency, some respondents were afraid that people in the U.S. would look down on them. Language barriers also made some older immigrants feel it was hard to integrate into the host society. P11 (Male, 73) stated that it was hard to integrate into the society because they don’t know English. P2 (Female, 74) also mentioned that they feel an ambivalent sense of belonging to American society, due to the limited communication capability in English. Other participants stated:

I was nervous because I knew nothing. I didn’t know the language and I was afraid people here would discriminate against me.(P13, Female, 76)

Sometimes I really like [the U.S.], but sometimes I don’t like it so much, I don’t always like it. Sometimes when I am in trouble, there is no help. And there is always the thought that I will be deceived.(P10, Male, 75)

There are different cultural values in Chinese and American society, which can contribute to both perceived and experienced social exclusion among Chinese American older adults. For example, based on the participants’ understanding or expectation, social relationships with neighbors are closer in China, while in the U.S. these relationships are more distant. This may make older Chinese immigrants feel it is difficult to integrate into the neighborhood. For example, P2 (Female, 74) mentioned, “I’ve been here for five years. Neighbors don’t interact with each other very much. Americans are like this”. Similarly, other respondents identified this:

I lived in three or four places [and my] relationship [with my neighbors] was only a greeting. … People come out of their homes at the same time. You don’t look at me and I don’t look at you. That is, an interpersonal relationship is the only greeting.(P12, Male, 68)

[Neighbors] They have their own life…do their own things, have small family…. This is American style…. You don’t bother people and they don’t bother you. Americans are like this, which is different from our Chinese values. People help and support each other in traditional Chinese value…a system of big family.(P21, Male, 68)

### 3.5. Exclusion from Neighbourhood/Territory/Community and the Risk Factors

Some participants reported territorial exclusion, such as a reduced geographic living area and an unsafe neighborhood. A few participants felt lucky to live in Los Angeles because of a high-density Chinese population. However, some participants also reported that they sometimes feel unsafe in their own neighborhood. For example, P2 (Female, 74) mentioned that the senior apartment she lived in used to be safe, but not anymore, because a TV set was stolen from a common room in the basement. Safety issues in the neighborhood were also reported by P18:

Sometimes, there are people who want to break in. There is no safe place nowadays. It depends…there is no safe place. Even now you go onto the street, older people need to be very careful when you go to the market to buy stuff. You need to carry your handbags on your shoulder, don’t lift it [in your hand]. Otherwise, bad guys will rob you. (P18, Female, 69)

The lack of safety in the neighborhood could also be related to racial discrimination, as perceived by one participant:

A batch of people lives nearby our house. They are Whites, so they look down on us. They sometimes throw some plates to my roof, climb up on our roof, and step on our house.(P10, Male, 75)

Exclusion from the neighborhood is also seen in older Chinese immigrants’ absence of a close relationship with neighbors. Participants feel excluded from the neighborhood they live in, especially when where there are a large number of Westerners:

Some Chinese people feel low self-esteem when they go to westerners’ neighborhood, or they feel different from them…I heard from my friends sometimes that if there is only one or two Chinese [in the neighborhood], they live very uncomfortably.(P17, Female, 65)

### 3.6. Summary of the Findings: Accumulating Disadvantages

Figure 1 summarizes the accumulating disadvantages of how aging-related and immigration-related changes can work together to contribute to the social exclusion experienced by the participants, which is clearly a reflection of the theoretical framework of old-age vulnerability. As seen in Figure 1, old age and immigration-related factors likely lead to five domains of social exclusion; the domain of civic participation was not mentioned by participants in the present study.

## 4. Discussion

In interviewing 24 older Chinese immigrants living in Los Angeles, the research findings showed that older Chinese immigrants experienced one set of multi-dimensional social exclusion faced by older adults, which is consistent with the literature. More importantly, this study revealed that, compared to the general older adult population, older Chinese immigrants face another set of immigration-related disadvantages, such as changes in their socio-cultural environment, physical environment, and legal status, which may aggravate the situation of being socially excluded (as illustrated below). This specific immigrant- related exclusion experience is similar to the experience of immigrants from other countries, such as the general immigrant population in Europe and African immigrants in the United States [22,23,24,25], as well as the older Korean immigrants in the United States [33].

### 4.1. Social Exclusion Experienced by Older Chinese Immigrants

Even though Chinese immigrants are usually considered to be financially well-off and taken good care of by their family members, our study showed that older Chinese immigrants experienced multi-level dimensions of social exclusion, reflected in various economic, social, and cultural aspects of their daily lives, including their access to basic service; material and financial resources; social relations and activities; socio-cultural aspects; and neighborhood/territory/community. All five of these dimensions of social exclusion were shown in the systematic review provided by Walsh et al. [17], except exclusion from civic engagement Although one participant mentioned that, in his opinion, most older Chinese American adults tended not to vote, most older adults interviewed did not report exclusion from civic engagement. It might be that civic engagement is not considered an important aspect of daily life by the Chinese immigrant population [49], especially among older Chinese immigrants, who were reported to have the lowest participation-rates in voting [50,51]. Another possible reason for this finding is that the participants were relatively less educated. Older adults with higher levels of education would be more likely to vote and value civic engagement [52].

Among the forms of social exclusion that older Chinese immigrants experienced, exclusion from family and social relationships was mentioned most frequently. Most older Chinese immigrants come to the U.S. to join their adult children, and view such reunification and connection with children as an important family tie [53]. They still emphasize traditional Chinese intergenerational interdependence after immigration, such as frequent contact with and greater reliance on children for practical support such as transportation, language brokering, help with personal care, and financial assistance [54,55]. However, the adult children are often too busy with their employment or their own nuclear family to provide the amount or quality of support or care that their aging parents expect. This finding does not indicate poor intergenerational support or contradict empirical quantitative studies that show that Chinese immigrants use intergenerational support and/or friends to cope with adversities in later life [41,56]. On the contrary, this qualitative study vividly depicts the desire not to be excluded from traditional family ties or support from children, which respondents would certainly have experienced had they not immigrated to the U.S., but which they realize can occur in the U.S. [41].

Beyond support from children and connections with friends, engagement in social activity in the neighborhood is a key indicator of successful aging among older adults in general [57]. Many older people have spent a substantial period of their lives in a particular neighborhood, have strong emotional investments in the surrounding community, and tend to increasingly rely on neighborhood relationships for support in old age [58]. Being involved in social activities in the host community is particularly important for older immigrants, because community activities can facilitate their settlement process. Participants indicated an interest in, and desire to participate in, social activities. However, they felt excluded through a lack of access to neighborhood or community services, as well as cultural and religious activities. This finding is consistent with previous study findings in older Chinese immigrants in Australia, who reported a desire to be actively involved in social events to eliminate feelings of loneliness and isolation [59]. This was also true in a study of older Korean immigrants who had experienced exclusion from social engagements [33]. 

Housing and transportation are important indicators that greatly influence the quality of life. Participants reported exclusion from such basic services. Older Chinese immigrant participants also felt exclusion from material resources. Without personal income and familiarity with American culture, even older immigrants who are former professionals may depend on their adult children for financial support [32,60]. Such financial- or material-resource shortages have been reported in other studies. For example, Korean Americans were found with no homeownership or retirement-pension ownership, limited work opportunities, and limited income from social security [33].

### 4.2. Accumulating Disadvantages of Aging and Immigration for Social Exclusion

Consistent with the “old-age vulnerability” framework, older immigrants’ risk of social exclusion resulted from the accumulating disadvantages of aging and immigration. Being old (state/status/exposure) makes people more vulnerable to social exclusion. As people get old, they inevitably experience a decline in physical health, limited function and mobility, loss of or reduced income, and a decrease in social connection, which may make them more vulnerable to social exclusion. These age-related factors of social exclusion among older Chinese immigrants were consistent with previous studies among other populations [16,37].

Immigration, as a specific event, may become a compounding threat for older adults (not necessarily for younger people), because older immigrants are more prone to, and have less resilience to, social exclusion. Most of the immigrants in this study appear to have immigrated in mid or later life, a life stage which presents unique problems for adaptation. At older ages, it is more difficult to learn a new language, adapt to a new culture, establish new social contacts, or find employment. The dramatic changes older Chinese immigrants experience in their economic, familial, and social structure can also exacerbate their vulnerability to social exclusion. As Figure 1 shows, changes in socio-cultural environment, physical environment, and legal status all contribute to dimensions of the social exclusion experience.

#### 4.2.1. Socio-Cultural Environment

This study found that immigration-related factors, changes in the socio-cultural environment, physical environment, and legal status, all contributed to dimensions of the social exclusion experience. The language barrier may be the biggest risk factor for feeling social exclusion, because it affects daily interactions with neighbors and community members, and the receiving of social/medical services from English-speaking providers. One study showed that older Chinese immigrants have a low level of acculturation, indicated in their pronounced preference for the daily language use of Chinese [27]. Their low proficiency in speaking the English language may prevent many members of this group from making social contacts and participating in community activities, which would contribute to older immigrants’ sense of belonging to the host society [41].

Furthermore, cultural values, norms, and the expectations of older Chinese adults also play a role in their experience of social exclusion. Older Chinese immigrants draw on attitudes from two different cultures: Eastern values of interdependency and Western values of independence/autonomy. Although living in a country with a dominant Western culture, these first-generation older Chinese immigrants may still have Eastern values deeply rooted in their psychology. Research has shown that, compared to other minority groups, Chinese Americans are more likely to carry their values, customs, lifestyle, and beliefs with them wherever they go, particularly first-generation immigrants [61].

The acculturation gap between adult children and older parents in this population is a potential reason for the experience of feeling excluded from family relationships [62]. With an expectation that they can be highly dependent on their adult children and will have a close relationship with their neighbors (i.e., more greetings or collective activities such as group morning exercise), just as they experienced in China before they came to the U.S., older Chinese immigrants will feel excluded when the family structure, intergenerational relationships, and social interactions change in host countries that deemphasize interdependence. Moreover, misunderstandings caused by cultural differences between the U.S. and China also partially contribute to this group’s sense of social exclusion from friendship networks or social contacts.

#### 4.2.2. Physical Environment

Immigration also results in a disruption of previous close networks in the home country. Older immigrants in general do not have many established social contacts or networks in the host country. Because of the changed family- and friend-network, the participants thus felt excluded from close relationships with family or friends. This finding is consistent with previous study findings for other minorities, which indicated that immigration and acculturation tend to disrupt socialization and the bonding process within family and friend networks [32,60,63]. For example, a study found that older Chinese and Korean immigrants in San Francisco felt that they became peripheral family members, were no longer authority figures in families, and were more independent, after immigration to the U.S. [64]. Relocating to a new neighborhood poses challenges for older immigrants, both in forming new relationships and feeling a sense of belonging. A racially diverse neighborhood can also exacerbate older immigrants’ feelings of being excluded.

#### 4.2.3. Immigration/Legal Status

Immigration status influences participants’ eligibility for using social services. Not holding U.S. citizenship makes some older Chinese immigrants feel excluded from material resources that would allow them to pay for medical services. Their immigration status also determines their eligibility for applying for or enjoying some social or medical services and their civic participation.

### 4.3. Study Limitations, Contributions, and Implications

There are some limitations for interpreting the findings of this study. First, this study was conducted in Los Angeles, which has a large Chinese/Asian population and more social support systems. Therefore, the findings from the present study did not represent those living in areas with much smaller Chinese/Asian populations and less social support. Second, due to the nature of qualitative data, the participants’ social exclusion experience was not differentiated by their demographic characteristics such as gender, age, length of stay, family status, and income. Quantitative research should be conducted in the future to address the heterogeneity of social exclusion and quantify its associated risk-factors among this population. Third, although the present study focused on exclusion from society, exclusion may include both self and social exclusion, which sometimes entangle together. Future study can distinguish between these two forms of exclusion and examine their relationship. Lastly, there are limitations in using relatively old data which do not contain social exclusion experiences related to COVID-19. Future study may consider collecting both qualitative and quantitative data on social exclusion related to COVID-19. 

Despite its limitations, this study contributes to social exclusion theory by applying it to older immigrant adults. The construct of social exclusion remains ambiguous in the context of aging populations [17,18,65]. However, social exclusion can offer valuable insights into the complexity of disadvantages affecting older immigrant individuals and groups. This study confirmed the multidimensional concept of social exclusion and also explored the unique risk factors for the social exclusion of older Chinese immigrants, aside from other common rick factors for the general older population. By applying “old-age-vulnerability theory” to immigrant older adults, this study found that older Chinese immigrants seemed to have few opportunities and pathways to lift themselves out of exclusion, due to the unique social, cultural, and immigration-related risk factors that are hard to modify. Thus, they are susceptible to the exclusionary processes intersecting their lives and are vulnerable to the impacts of such exclusion mechanisms. To fill a major gap in the literature, this study attempts to elaborate the potential pathways to social exclusion faced by older immigrants, which can provide some operational guidelines for future studies on this population. Moreover, considering the accumulating disadvantages of aging-related and immigration-related factors can lend a framework to studies on other marginalized groups of older people, such as LGBTQ individuals or those with disabilities. 

Furthermore, findings from this study have important implications for practice and service provision for older Chinese immigrants. First, practitioners need to be educated about social exclusion and its manifestations in daily life within this population. They need to be aware of the challenges older Chinese immigrants may face, and understand “the model minority” myth that may reduce the social inclusion effort made for this population. Practitioners and providers should be prepared to use culturally informed and evidence-based interventions to promote their social inclusion and integration as well as health and well-being. However, the inclusion and integration of the older immigrant population into society is not the only solution to social exclusion. Society as a whole should also take actions to eliminate other contributing factors to social exclusion, such as discrimination or ageism, to help decrease the social exclusion experience of immigrant populations. Second, human service agencies, such as the Department of Health and Human Services, the Area Agency on Aging, adult day care centers, and senior apartments, could develop services or programs to help older adults meet their basic needs, such as transportation, housing, or medical services and referrals to affordable services in the community, to help this group achieve feelings of social inclusion. Given the language barriers many older adults in this group face, programs such as volunteer programs or intergenerational programs have the potential to help those older adults gain some basic English-language skills and potentially help them make some new connections with people. Third, since the U.S. is one of the major destination countries for migrant people, the outcome of this study on older Chinese immigrants in the U.S. may contribute to future integration-policy development, so that older immigrants will not be excluded from support and development initiatives for ensured access to basic services, effective communication with relatives, social relationships, sensitivity to cultural aspects, and support for community issues.

## 5. Conclusions

Social exclusion offers the potential to understand the life course features of the disadvantages and cumulative inequalities that occur in exclusionary mechanisms among older Chinese immigrants. A specific focus on immigrants’ old-age exclusion may offer a valuable approach for informing and evaluating age-related social policy for immigrants in the United States. It is also likely to be particularly relevant, given the prevailing economic austerity in the U.S., and the potential for austerity to increase older people’s exclusion [17]. This study provides useful information and strategies for human and healthcare service providers to find ways to overcome social exclusion and enhance older Chinese immigrants’ social inclusion in the U.S. It also contributes to social exclusion theory by calling for more attention to specific older adult groups that face disadvantages from multiple sources.

## Figures and Tables

**Figure 1 ijerph-20-02539-f001:**
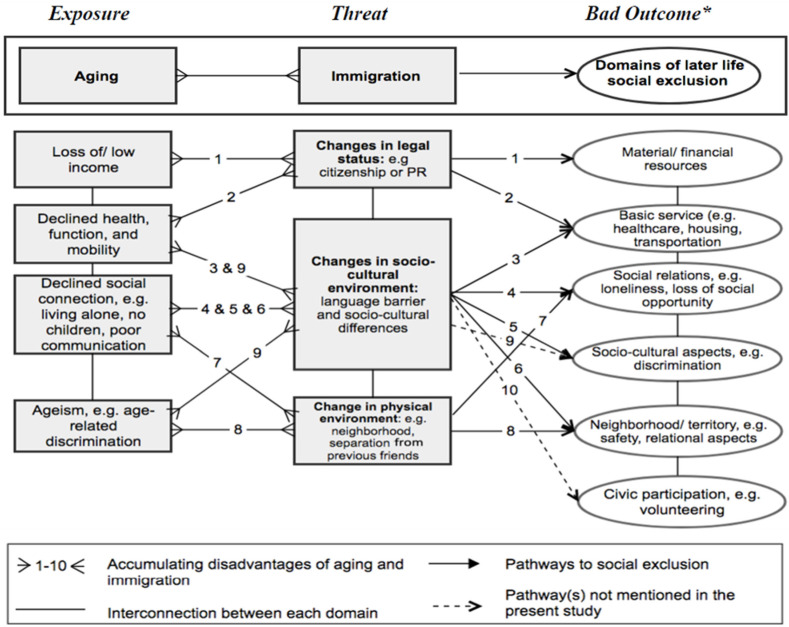
Pathways to social exclusion experienced by older Chinese immigrants. * Adaptation can be those positive approaches for achieving/maintaining social inclusion by avoiding social exclusion. Note: PR is permanent residence.

**Table 1 ijerph-20-02539-t001:** Demographic characteristics of participants (*n* = 24).

Variables	*n*	Mean (SD)/%	Range
Gender (female)	13	54.2	
Age	24	76.9 (7.5)	65–92
Marital status			
Married	15	62.5	
Widow	7	29.2	
Divorced	2	8.3	
Education			
No Education	3	12.5	
Elementary	4	16.7	
Middle School	4	16.7	
High School	5	20.8	
College or above	8	33.4	
Religion (yes)	8	33.3	
Work status (still working)	6	25.0	
Living arrangement			
Alone	9	37.5	
Only with spouse	10	41.7	
Only with children	3	12.5	
With children and spouse	1	4.2	
With friends	1	4.2	
Citizenship			
Naturalized citizen	21	87.5	
Permanent resident card holder	3	12.5	
Years in the U.S.	24	21.9 (12.8)	1–44
Reasons for immigration			
Family reunion	18	75.0	
Education	1	4.2	
Others ^a^	5	20.8	
Self-Identity			
American	3	12.5	
Chinese	9	37.5	
Chinese American	10	41.7	
Chinese and American	2	8.3	
Language speak at home			
Mandarin	11	45.8	
Cantonese	13	54.2	
Speak English (yes)	17	70.8	

^a^ Due to political concerns, to improve prospects for their children, etc.

## Data Availability

This research did not use a publicly available dataset.

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
