# Peer review of "Exploration of Social Exclusion among Older Chinese Immigrants in the USA"

_ijerph, 2023, doi:10.3390/ijerph20032539_

Round 1

Reviewer 1 Report

Thank you for giving me an opportunity to review this interesting paper regarding the social exclusion issue among Older Chinese Immigrants in the United States of America. Based on 24 face-to-face individual interviews conducted in Los Angeles in 2010, the authors pointed out  that older Chinese immigrants experienced multi-dimensional social exclusions. The article lays out its argument clearly. Nevertheless, the marginal contributions might be mainly on the descriptive level. There are some problems that the authors should carefully address:

1. As a qualitative study, this paper lacked of a consistent theoretical framework. Although the authors used the old-age vulnerability framework including such domains as exposure, threats, coping capacities and bad outcomes, the demonstration of actual research findings did not strictly refer to this framework. Instead, Walsh’s framework was harnessed to describe exclusion related to basic service, material and financial resources, social relations and activities, socio-cultural aspects of society, and neighborhood/territory/community. The authors need to provide necessary explanations about the connections and differences between these two analytical frameworks.

2. The reasons for the selection of 24 cases were not specified clearly. Compared to native residents and immigrants from other countries, are there significant differences in the exclusion experiences among older Chinese immigrants?

3. As mentioned in the paper, old-age vulnerability framework included coping strategies to overcome adverse experience. However, the research results were mainly concerned with manifestations of the social exclusion problem that older Chinese immigrants faced. It is necessary to supplement the qualitative interview contents related to the older Chinese immigrants’ coping strategies during the process of experiencing social exclusion problem.

In summary, although the topic of this manuscript is attractive, the paper requires a Minor Revision to beef up its theoretical framing and explanation of results etc.

Author Response

  1. As a qualitative study, this paper lacked of a consistent theoretical framework. Although the authors used the old-age vulnerability framework including such domains as exposure, threats, coping capacities and bad outcomes, the demonstration of actual research findings did not strictly refer to this framework. Instead, Walsh’s framework was harnessed to describe exclusion related to basic service, material and financial resources, social relations and activities, socio-cultural aspects of society, and neighborhood/territory/community. The authors need to provide necessary explanations about the connections and differences between these two analytical frameworks.

Response: Thank you for the comments. The old-age vulnerability framework justifies the necessity and significance of this study to explore older immigrants’ experiences of social exclusion because this group faces double exposure and threats upon aging and immigration, which is worth studying. It also guides our exploration on potential risk factors of older immigrants’ social exclusion. On the other hand, Walsh’s framework helps us to categorize experiences of social exclusion into different domains. While the old-age vulnerability framework illustrates a potential pathway of where and how older Chinese immigrants’ experienced social exclusion may have come from and been shaped, Walsh’s framework helps us to summarize our qualitative findings and describe what they actually experienced in terms of social exclusion. We added explanations about the two frameworks in the revised manuscript.

  1. The reasons for the selection of 24 cases were not specified clearly. Compared to native residents and immigrants from other countries, are there significant differences in the exclusion experiences among older Chinese immigrants?

Response: The reasons for the final sample size (n = 24) was mentioned on page 5 (line 206-207) as “Sampling and interviewing were stopped when data reached saturation, with no additional data found to develop new categories or themes”.

The comparison of the present study findings with prior literature was shown in the discussion part (pages 12-14). In summary, compared to the general older adult population, older Chinese immigrants in the present study experienced similar social exclusion, such as the five dimensions of social exclusion by Walsh et al., except exclusion from civic engagement (page 12, line 491-493). However, older Chinese immigrants also face another set of immigration-related disadvantages, such as changes in their socio-cultural environment, physical environment, and legal status, that may aggravate the situation of being socially excluded (page 11, line 478-481). This exclusion experience is similar to the experience of immigrants from other countries, such as the general immigrants in Europe and African immigrants the Unites States [22-25]*, as well as the older Korean American immigrants [33]*. We added this comparison summary of the present study with immigrants from other country at the bottom of page 11. Other comparisons between the present study and prior literature on each dimension of social exclusion were discussed on page 12-14.

*Citations can be found in the main text.

  1. As mentioned in the paper, old-age vulnerability framework included coping strategies to overcome adverse experience. However, the research results were mainly concerned with manifestations of the social exclusion problem that older Chinese immigrants faced. It is necessary to supplement the qualitative interview contents related to the older Chinese immigrants’ coping strategies during the process of experiencing social exclusion problem.

Response: We appreciate this suggestion from the reviewer. We agreed with the reviewer that exposure, risk, and coping capacity work together to result in social exclusion based on the theoretical framework of the old age vulnerability. As we mention in the text (page 4, line 150-154), “Therefore, older Chinese immigrants are more vulnerable to social exclusion than general older adults due to the emerging threats from unfamiliar physical, social and cultural environment, especially when the coping capacity of older immigrants themselves, their families, and their host society cannot mitigate these threats”. We will consider conducting another qualitative interview with contents of coping strategies during the process of experiencing social exclusion problems.

Reviewer 2 Report

Based on in-depth interviews with 24 older Chinese immigrants residing in Los Angeles, this study identified five dimensions of social exclusion experienced by participants. This project is socially significant. The research method is valid and rigorous. The findings provide a valuable addition to the literature. The manuscript was very well-written, informative, and well-organized. I appreciate the authors for their hard work and contribution to the literature. I only have a few minor suggestions and comments.

1. In the introduction, the authors mentioned: “the age of the data” (Lines 49-50). While I don’t think this is an issue and I believe the authors did a great job justifying the use of the older data, I suggest the authors be more explicit and clearly mention that this study was based on 2010 data. The current wording may leave readers the impression that the authors try to hide something. 

2. In Section 2.2. Procedures, I’d suggest the authors clearly mention that the interviews were conducted in Chinses. 

3. In Section 2.3. Ethics, since this is a single-blind review, I think the authors should specify which university’s IRB approves the current study. 

4. The authors did a great job presenting the results. Figure 1 is particularly informative and useful. 

5. Another limitation is that this study was conducted in LA, which has a large Chinese/Asian population and more social support systems. The authors may mention how the findings did not represent those living in areas with much fewer Chinese/Asian populations and social support. 

Author Response

  1. In the introduction, the authors mentioned: “the age of the data” (Lines 49-50). While I don’t think this is an issue and I believe the authors did a great job justifying the use of the older data, I suggest the authors be more explicit and clearly mention that this study was based on 2010 data. The current wording may leave readers the impression that the authors try to hide something.

Response: Thanks for this suggestion. We changed this sentence to “despite the age of the data used in the present study that were collected in 2010” to be more explicit as suggested.

  1. In Section 2.2. Procedures, I’d suggest the authors clearly mention that the interviews were conducted in Chinses.

Response: We added this sentence as suggested (on page 6, line 219).

  1. In Section 2.3. Ethics, since this is a single-blind review, I think the authors should specify which university’s IRB approves the current study.

Response: We added the university name in this revised manuscript.

  1. The authors did a great job presenting the results. Figure 1 is particularly informative and useful.

Response: Thank the reviewer for this encouraging comment.

  1. Another limitation is that this study was conducted in LA, which has a large Chinese/Asian population and more social support systems. The authors may mention how the findings did not represent those living in areas with much fewer Chinese/Asian populations and social support.

Response: Thanks for this comment. We added this limitation on page 14 in the revised manuscript.